# Possible Brainstem Anaesthesia in a Cat after Ultrasound-Guided Retrobulbar Block

**DOI:** 10.3390/ani13050781

**Published:** 2023-02-21

**Authors:** Anastasia Papastefanou, Eva Rioja

**Affiliations:** Optivet Referrals, Ltd., Havant PO9 2NJ, UK

**Keywords:** brainstem anaesthesia, cat, retrobulbar block, intraarticular injection

## Abstract

**Simple Summary:**

Very commonly, local anaesthetic is deposited behind the eye to provide analgesia in animals that have an eye removed. In humans, this technique has been associated with complications like seizures and apnoea, while this has been reported only once, so far, in a cat. We present a case report of acute respiratory arrest during anaesthesia that occurred after local anaesthetic was administered behind the eye of a 13-year-old Siamese cat under ultrasound guidance. The cat stopped breathing almost instantly, and its heart rate and blood pressure increased significantly. The high blood pressure resolved quite quickly, but the cat very quickly developed hypotension and bradycardia, which were difficult to treat. Spontaneous breathing returned 20 min after the end of anaesthesia. The cat recovered well, but the contralateral eye did not respond to light and vision and seemed to be impaired. The spread of the local anaesthetic in the brain of the cat was suspected, based on the similarities with other reported cases, mainly in humans. The following day, the symptoms improved but were not completely resolved. One week later, during a follow-up examination, there were no concerns. The addition of this case report to the literature will help to raise awareness regarding this rare but potentially life-threatening complication.

**Abstract:**

A 13-year-old neutered, blue-eyed female Siamese cat with a bodyweight of 4.8 kg was admitted for enucleation of the right eye. An ultrasound guided retrobulbar block with 1 mL of ropivacaine was performed under general anaesthesia. When the tip of the needle was visualised inside the intraconal space, negative aspiration of the syringe before injection and no obvious resistance during injection were confirmed. Instantly, after ropivacaine was administered, the cat became apnoeic, and its heart rate and the blood pressure increased significantly for a short period of time. During surgery, the cat needed cardiovascular support to maintain blood pressure and was under continuous mechanical ventilation. Spontaneous breathing returned 20 min after the end of anaesthesia. Brainstem anaesthesia was suspected, and after recovery, the contralateral eye was examined. A reduced menace response, horizontal nystagmus, mydriasis, and absence of the pupillary light reflex were present. The following day, mydriasis was still present, but the cat was visual and was discharged. The inadvertent intra-arterial injection of ropivacaine was suspected to be the cause of the spread into the brainstem. To the current authors’ knowledge, possible brainstem anaesthesia has only been reported in a cat 5 min after a retrobulbar block but never instantly.

## 1. Case Presentation

A 13-year-old neutered, blue-eyed female Siamese cat with a bodyweight of 4.8 kg was scheduled for enucleation of the right eye due to panuveitis and secondary glaucoma. The menace response and dazzle reflex were normal on the left eye. Apart from being geriatric, the cat had no other comorbidities in her clinical history. Her preoperative clinical examination was normal without any nystagmus noticed or reported during her ophthalmological examination, and her biochemistry and haematological results were unremarkable. The cat’s current medications included meloxicam and chloramphenicol ointment. For preanaesthetic medication, a combination of methadone (0.2 mg/kg) (Methadyne, Jurox, Leatherhead, UK), midazolam (0.3 mg/kg) (Hameln pharma Ltd., Gloucester, UK), and alfaxalone (1.5 mg/kg) (Alfaxan/Multidose, Jurox, Dublin, Ireland) was mixed in the same syringe and administered intramuscularly (IM). Anaesthesia was induced with a total of 1.8 mg/kg of alfaxalone administered through a right cephalic intravenous cannula. Intravenous fluid therapy was administered during anaesthesia at a rate of 3 mL/kg/h (Hartmann’s solution; Aquapharm No. 11; Animalcare LTD; York, UK). Anaesthesia was maintained with sevoflurane (Sevohale, Chanelle, Galway, Ireland) in 100% oxygen administered through a Mapleson-D breathing system. Throughout anaesthesia, heart rate and rhythm with electrocardiography, haemoglobin oxygen saturation (SpO_2_), oscillometric blood pressure, oesophageal temperature, capnography, and spirometry were continuously monitored and recorded every 5 min using a multiparameter monitor (Datex-Ohmeda, GE, Helsinki, Finland). A retrobulbar block (RBB) was used as part of a multimodal analgesic plan. Following aseptic preparation of the periocular area and the area dorsal to the zygomatic process, an ultrasound (US)-guided RBB was performed using a 14 MHz linear transducer connected to a portable ultrasound device (Mindray bio-medical electronics Ltd., Shenzhen, China). Sterile ocular gel was applied to the globe, and the US transducer was placed over the cornea with the marker positioned laterally. The orientation was initially perpendicular to the transverse plane and parallel to the dorsal plane of the globe, and then it was oriented with an angle of 30° approximately relative to the dorsal plane in order to position the US marker towards the junction between the zygomatic arch and the orbital ligament laterally. The depth and focus of the US were adjusted to achieve a good view of the retrobulbar space, the cone, and the optic nerve. The retrobulbar compartment was identified as a conic shape formed by the extraocular muscles positioned caudally to the eyeball. A 22 gauge, 50-mm ultrasound needle with extension tubing (USB 50 EVOLUTION, Temena GmbH) was introduced using a supra-temporal approach [1]. The needle was inserted perpendicularly to the skin and oriented latero-medially behind the orbital ligament, dorsally to the zygomatic process of the temporal bone and ventral to the frontal bone, through the temporal muscle. When the tip and part of the needle were identified ultrasonographically using an oblique in-plane approach, the needle was advanced latero-medially to reach the intraconal space by crossing the extraocular muscles. In this case, the needle was visualised to go through the cone and appear in the medial aspect of it; therefore, it was slightly retracted until it was inside the cone again. At this level, negative pressure was applied to the syringe to exclude blood vessel penetration, and 1 mL of ropivacaine 0.75% (Naropin, Apen Pharma Trading Limited 0.75%, Dublin, Ireland) was injected without obvious resistance. The maximum recommended dose of ropivacaine of 2 mg/kg was calculated to be 1.28 mL [2].

Immediately after the injection and before the needle had been removed, the cat became apnoeic. After a minute of apnoea, manual ventilation was initiated. At the same time, there was an immediate rise in the cat’s heart rate (HR) from 102 to 140 bpm with a sinus rhythm. Unfortunately, there is no record of the concurrent blood pressure (BP), but we recall that it increased in parallel with the heart rate. Twenty minutes later, the cat was transferred to the operating theatre, and continuous mechanical ventilation (CMV) on a volume control mode was initiated using a veterinary ventilator (Merlin, Vetronic Services LTD, Devon, UK). During this time, no parameters were recorded on the monitoring sheet as the anaesthetist was involved in providing manual ventilation, but when the cat was moved to theatre, the blood pressure could not be measured with oscillometry or the flow Doppler technique. By that time, it was suspected that the administration of the local anaesthetic (LA) in the retrobulbar space could have been responsible for the occurrence of apnοea and the instability of the HR and BP. When the surgery started, the mean BP (MBP) was 50 mmHg, and the HR was 75 bpm. A dose of 0.01 mg/kg of glycopyronium (Accord Healthcare Limited, Durham, UK) was administered intravenously (IV) and was repeated after 5 min as there was no effect. Due to the lack of response to the anticholinergic, a dose of 0.1 mg/kg of ephedrine (MaCarthys Laboratories, Martindale Pharma, Brentwood, UK) was administered IV, and this was repeated again after 10 min, after which the MBP increased to 80 mmHg but only temporarily. The HR remained at levels >100 bpm after the above interventions, while the MBP increased again only after a dopamine (Martindale Pharma, UK) IV infusion at a dose of 5 μg/kg/min was initiated. During the whole anaesthetic, the SpO_2_ and the EtCO_2_ were maintained within the acceptable limits, while the oesophageal temperature was 34–36 °C. Anaesthesia ended at 75 min after induction, and the cat remained apnoeic. Twenty minutes following inhalant discontinuation, the cat was still apnoeic, and a decision was made to reverse the midazolam with 0.01 mg/kg of flumazenil IV (Hamlen Pharma gmbh, Gloucester, UK). The cat responded immediately and started breathing spontaneously following the administration of flumazenil. Dopamine was discontinued after extubation, as the cat was able to maintain MBP > 80 mmHg. During surgery, the cat lost approximately 29 mL of blood, which was 10% of her estimated total blood volume (4.8 kg × 60 mL/kg = 288 mL). On recovery, the cat remained on Hartman’s solution at a rate of 3 mL/kg/h until the volume lost was replaced. After the cat had fully recovered, an ophthalmological examination of the left eye revealed a reduced menace response, horizontal nystagmus, mydriasis, and absence of the pupillary light reflex (PLR). The rest of the clinical examination was normal. She was responsive to stimulation and had a good appetite. The Glasgow Feline Composite Measure Pain Scale scores were between 0 and 2 postoperatively. The following day, she had a normal menace response, subtle horizontal nystagmus, mydriasis, and the PLR was present. The cat returned home on the day after surgery. One week later, at the follow-up examination, the left eye had a normal menace response, no nystagmus, a slightly larger than normal resting pupil size but was less mydriatic than postoperatively and within the normal limits for a blue-eyed Siamese cat, and with the PLR present. The owner reported no concerns, and the patient was discharged.

## 2. Discussion

Regional anaesthesia of the eye is commonly applied in veterinary patients to provide analgesia during painful procedures, such as enucleation, or for akinesia of the extraocular muscles, which results in the globe maintaining a central position [3]. Additionally, it reduces the incidence of the oculocardiac reflex during enucleation surgery by blocking the ophthalmic branch of the trigeminal nerve and the ciliary ganglion, as has been described in horses [4]. The RBB and peribulbar blocks (PBB) are two different types of regional anaesthesia techniques in which the LA is either deposited in the retrobulbar or in the peribulbar space, respectively, with the final aim being for the LA to reach the intraconal space. The intraconal space is within the retrobulbar cone, and in the dog, the latter is formed by the ocular extrinsic muscles: the dorsal, ventral, lateral, and medial rectus muscles, the retractor bulbi with its four fascicles, and more superficially and medially, the dorsal oblique muscle [5]. There is no fascia surrounding the muscles that form the cone. Within the retrobulbar cone are the ophthalmic nerve, the ciliary nerves, the optic nerve, the oculomotor nerve, and the abducens nerves. Additionally, the internal ophthalmic artery and the ciliary arteries run in the same space [5].

Different techniques have been described for both blocks [2]. In dogs, among other blind approaches to the RBB, the inferior-temporal-palpebral [6] and the supra-temporal [1] approaches have been described in cadavers. In one study with sedated dogs, the peribulbar technique was shown to be more reliable for producing anaesthesia than the RBB when using a bent needle inserted through the inferior eyelid [7]. However, both methods produced adverse effects such as exophthalmos, chemosis, and anterior uveitis, which had resolved 14 h later [7].

In cats, Shilo-Benzamini et al. [8] described a superior-nasal approach for the retrobulbar block through the upper eyelid at the dorsomedial orbit using a curved needle. This was found to have a 70% success rate in cadavers, but this was only 50% when tested in sedated cats [9].

Ultrasound-guided ophthalmic regional anaesthesia has been used recently in human ophthalmology, as it has the advantage of allowing visualisation of the retrobulbar structures and the needle at the same time as well as confirmation that the LA has been administered in the correct space. According to our knowledge, thus far, only extraconal deposition of the LA has been described in dog cadavers with US guidance [10]. An US-guided PBB was shown to be as effective as a blind technique in anaesthetised dogs, although the dogs receiving the blind technique had a higher incidence of increased intraocular pressure (IOP) due to the possible wrong positioning of the needle, as the authors hypothesised [11].

In cats, an US-guided RBB with a trans-palpebral approach, as mentioned above, was effective in only 50% of the animals [9], possibly due to “the difficulty in identifying the curved needle tip”, as commented by the author in a review article on regional ophthalmic anaesthesia [2]. More recently, the authors of a case report on a cat with microphthalmia described the use of an US-guided RBB using a modified technique similar to the supra-temporal approach, but in this case, the needle was introduced into the retrobulbar space from the subzygomatic area [12].

Advocates of the PBB highlight the reduced risk of complications, such as the distribution of the LA in the central nervous system (CNS), retrobulbar haemorrhage, and globe penetration, compared to the RBB, as the needle is inserted further away from these structures [11]. Indeed, the reported incidences of CNS complications in humans after a RBB are 0.09%, 0.27%, and 0.9% [13,14,15], whereas a much lower incidence of 0.007% has been reported with the PBB; however, the authors concluded that this incidence is probably underestimated [16]. In veterinary patients, the incidence of complications is unknown. Only one case has been reported in a cat with suspected brainstem anaesthesia that occurred 5 min following the injection of LA into the retrobulbar space using a blind technique [17].

The technique routinely performed at our hospital is the US-guided RBB using a supra-temporal approach with intraconal deposition of the LA. This preference is due to the perceived greater success rates when an US-guided technique with an in-plane approach is used and the correct deposition of the LA can be visualised. Additionally, a smaller volume of LA is necessary, as it is injected inside the cone.

Humans receive regional ophthalmic blocks while awake, and the side effects vary from contralateral pupil dilation to amaurosis, convulsions, grand-mal seizures, hypertension/hypotension, tachycardia/bradycardia, and cardiopulmonary arrest [18]. In veterinary medicine, the side effects reported in one cat following RBB were apnoea, increased HR and systolic blood pressure initially, and a reduction in blood pressure below normal values during surgery [17]. A delay in recovery was also noticed, and after extubation, the cat presented with nystagmus, absence of the menace response, mydriasis, a lack of dazzle, and a negative PLR in the contralateral eye. There are three main potential mechanisms whereby a RBB or PBB may lead to CNS complications. First, these can occur due to penetration of the sheath of the optic nerve and the entry of LA into the subarachnoid space of the brainstem [14]. If this occurs, symptoms appear 5 to 10 min after injection, depending on the onset of action of the LA. The second mechanism is through the inadvertent injection of the LA into the internal ophthalmic artery. The pressure of the injectate forces the LA to flow back into the internal carotid artery and into the brain. In such cases, clinical signs appear rapidly [19]. Thirdly, complications can occur due to the systemic absorption of the LA [18].

In the case presented here, tachycardia, hypertension, and apnoea appeared instantly before the removal of the needle, and the cat remained mechanically ventilated until recovery. No obvious resistance was noticed during the injection, and the needle did not seem to have penetrated the optic nerve on the US image. Given the rapid onset of the clinical signs, CNS spread of the local anaesthetic exclusively through penetration of the sheath of the optic nerve is less likely. Unfortunately, there are no saved images to corroborate this, as it is not common practice to save images for every patient, especially in routine cases. Similar to our case and to the other case report from a cat, the first manifestations of CNS spread in humans are often hypertension and tachycardia. This can be explained as a parasympathetic blockade through a combined vagal and carotid sinus reflex block [20]. Although the immediate onset of symptoms in our case is in favour of inadvertent intra-articular injection, no blood appeared during the application of negative pressure to the syringe. Additionally, one would expect the duration of the symptoms to be transient due to the rapid redistribution of blood out of the brain, but this was only the case for hypertension and tachycardia. However, the cat remained apnoeic during the whole anaesthetic procedure and for 20 min after the end of anaesthesia. In dogs, the internal ophthalmic artery runs on the dorsal surface of the optic nerve [4]. To the best of our knowledge, there is no published description of the ophthalmic vasculature in cats that mentions the exact pathway of the internal ophthalmic artery. In humans, because of an anatomical variation in the inferior ophthalmic artery in 15% of the population, there is a higher risk of inadvertent intra-arterial injection [18]. The use of colour-flow Doppler could have helped us to identify the artery and avoid a possible intra-arterial injection of the LA, but this was omitted before the administration of ropivacaine. The rapid systemic absorption through the local capillaries could also explain this clinical presentation. However, the total dose administered (1 mL) was less than the maximum recommended dose in cats [2,21]. The possibility that some of the LA was injected intra-arterially and some reached the subarachnoid space through penetration of the sheath could explain the quick onset of symptoms in combination with the increased duration of apnoea. In a similar case report in a human, an intra-arterial injection was suspected due to the instant occurrence of CNS toxicity. The same patient remained apnoeic for 40 min after seizures had resolved. The authors speculated that systemic absorption from the local capillaries or the spread of the LA in the subdural space along with the optic nerve could explain the sustained respiratory arrest [18].

Following recovery from anaesthesia, the cat in the present case report had symptoms in the contralateral eye that could potentially be explained by brainstem exposure to the LA [14]. Although nystagmus may be present during recovery from anaesthesia and opioids cause mydriasis in cats, the above symptoms were still present the following morning, although to a lesser degree. In contrast with humans, where convulsions or seizures are common, the cat in this report did not show any twitches, probably because general anaesthesia was masking this response.

It is not clear why, upon administration of flumazenil, the cat started breathing spontaneously immediately, as midazolam causes minimal respiratory depression, and it did not compromise ventilation after administration in our case. However, when diazepam is administered with buprenorphine in rats, respiratory depression has been reported to be greater [22]. Potentially, the impact of benzodiazepines on central ventilatory control after CNS dysfunction, as occurred in our case, is more pronounced compared with in a normal CNS. Additionally, the use of flumazenil has been reported to reduce the recovery time from anaesthesia in humans when no benzodiazepines have been administered [23,24]. It is unknown as to whether it was the pharmacological action of flumazenil or the time that had elapsed since the retrobulbar block that caused the return to spontaneous breathing.

Lipid emulsion therapy has been proven to be effective for treating local-anaesthetic-induced systemic toxicity [25], including a case of local-anaesthetic-induced seizures after a peribulbar block in humans [26]. When brainstem anaesthesia was suspected in our case, the administration of a lipid emulsion could have potentially helped to resolve the cardiovascular and respiratory complications.

## 3. Conclusions

We have presented a case report of a potential exposure of the CNS to ropivacaine after an US-guided RBB in a cat that showed sustained apnoea, cardiovascular instability, as well as nystagmus, mydriasis, and an absent PLR postoperatively in the contralateral eye. The early onset of symptoms, prior to needle withdrawal, indicates that an inadvertent intra-arterial injection of the LA was the possible cause, although we cannot exclude subarachnoid injection or a combination of the two. Although an US-guided technique was used and negative pressure was applied before the injection, this complication could not be avoided, and we highlight the importance of using colour-flow Doppler for better visualisation of the vasculature. Any medications that can be reversed could be beneficial for similar cases.

## Data Availability

The presented case report is not a study, and no data were collected.

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
