# Peer review of "Possible Brainstem Anaesthesia in a Cat after Ultrasound-Guided Retrobulbar Block"

_animals, 2023, doi:10.3390/ani13050781_

Round 1
Reviewer 1 Report
Dear authors, thank you for this very well described case.
English editing should be further used as I found some sentences that I felt were missing some words or had a structure needing improvement.
An example in the simple summary ligne 13-14: "The high blood pressure was resolved quite quickly but developed hypotension and bradycardia, that were difficult to treat.
Somme grammar error are also identifyable. An example lines 45-47: a comination of .... was and note were.
Line 70: the sentence starts with the site of injection described asperpendicular to the skin. Would it rather be the needle orientation?
Line 79: more appropriate to describe the ropvacaine dose as recommanded rather than allowed. A refernec for this staement wpuld also be welcome.
Line 243, missing verb in the last sentence?
Concerning the case description that is well done, I have a few suggestions/questions:
- Was the heart rhythm of sinusal origin event during the tachycardic period? Eventhough, the dose of ropivacaine was lower than the recommanded dose and the chosen local anesthetic drug was less likely to be cardiotoxic, ventricular tachyarrhythmias have been described with systemic effects of LA and also in case of brainstem dysfonction.
- Blood pressure support is well described except for the duration it was needed and the time of discontinuation respective to the end of anesthesia and the early recovery period.
- Concerning the neurologic exam, nystagmus being a possible congenital abnormality in Siamese, it is worth mentioning that it was not identified during the preoperative examination.
- Your discussion brings really good points. I would suggest you go further into developping the apnea resolving after fumazenil administration. Although benzodiazepines are advocated to have very little effect on respiratory depression, clinically CNS depression can be more profound than expected especially with some degree of CNS dysfonction. It has been shown in humans that it has some degree of respiratory depression when used with opioids. Here is a reference in rats: https://www.bjanaesthesia.org.uk/article/S0007-0912(21)00701-7/fulltext.
Moreover, the use of flumazenil at the end of anesthesia has also been reported to hasten awakening in people even without prior benzodiazepine administration (https://www.ncbi.nlm.nih.gov/pmc/articles/PMC3272523/, https://www.ncbi.nlm.nih.gov/pmc/articles/PMC5326685/pdf/nihms836308.pdf).
- Finally, Iv lipid emulsion have been used to treat local anesthetic toxicity including brainstem anesthesia (here is a cas report : https://www.researchgate.net/publication/284217684_Successful_lipid_rescue_of_local_anesthetic_systemic_toxicity_following_peribulbar_block)
Developing the discussion on those two apsects would bring a nice conclusion and opening for your case.
Best regards
Author Response
Thank you very much for the constructive comments.
Please find our response to your comments written in Italics.
Dear authors, thank you for this very well described case.
English editing should be further used as I found some sentences that I felt were missing some words or had a structure needing improvement.
An example in the simple summary ligne 13-14: "The high blood pressure was resolved quite quickly but developed hypotension and bradycardia, that were difficult to treat.
Thank you, we have amended the sentence.
Somme grammar error are also identifyable. An example lines 45-47: a comination of .... was and note were.
Line 70: the sentence starts with the site of injection described asperpendicular to the skin. Would it rather be the needle orientation?
Please see the changes in the manuscript.
Line 79: more appropriate to describe the ropvacaine dose as recommanded rather than allowed. A refernec for this staement wpuld also be welcome.
It has been added.
Line 243, missing verb in the last sentence?
Has been amended.
Concerning the case description that is well done, I have a few suggestions/questions:
- Was the heart rhythm of sinusal origin event during the tachycardic period? Eventhough, the dose of ropivacaine was lower than the recommanded dose and the chosen local anesthetic drug was less likely to be cardiotoxic, ventricular tachyarrhythmias have been described with systemic effects of LA and also in case of brainstem dysfonction.
In our case, it was sinus tachycardia. It has now been added to the manuscript.
- Blood pressure support is well described except for the duration it was needed and the time of discontinuation respective to the end of anesthesia and the early recovery period.
It has been added to the manuscript.
- Concerning the neurologic exam, nystagmus being a possible congenital abnormality in Siamese, it is worth mentioning that it was not identified during the preoperative examination.
Thank you for pointing this out. There was no report that the cat had any degree of nystagmus previously.
- Your discussion brings really good points. I would suggest you go further into developping the apnea resolving after fumazenil administration. Although benzodiazepines are advocated to have very little effect on respiratory depression, clinically CNS depression can be more profound than expected especially with some degree of CNS dysfonction. It has been shown in humans that it has some degree of respiratory depression when used with opioids. Here is a reference in rats: https://www.bjanaesthesia.org.uk/article/S0007-0912(21)00701-7/fulltext.
Moreover, the use of flumazenil at the end of anesthesia has also been reported to hasten awakening in people even without prior benzodiazepine administration (https://www.ncbi.nlm.nih.gov/pmc/articles/PMC3272523/, https://www.ncbi.nlm.nih.gov/pmc/articles/PMC5326685/pdf/nihms836308.pdf).
We believe that is not very clear how midazolam affects respiratory function. Although it is commonly accepted that respiratory depression is not significant, we do tend to use it when an animal is fighting the ventilator and we want to control the ventilation mechanically. The fact that the second reviewer has a completely opposite opinion regarding the effect of midazolam on the respiratory system highlights the confusion. However, we have now added the recommended references and added more comments in the discussion about benzodiazepines and flumazenil's effect on respiratory function.
- Finally, Iv lipid emulsion have been used to treat local anesthetic toxicity including brainstem anesthesia (here is a cas report : https://www.researchgate.net/publication/284217684_Successful_lipid_rescue_of_local_anesthetic_systemic_toxicity_following_peribulbar_block)
Thank you for mentioning the lipid emulsions for local anaesthetic toxicity treatment. We have now added it to the manuscript.
Developing the discussion on those two apsects would bring a nice conclusion and opening for your case.
Best regards
Reviewer 2 Report
This is an interesting case report about the potential brainstem anaesthesia secondary to retrobulbar block in a cat, with a supra-temporal approach. The description of the case is extensive and clear. Although this complication has been described previously, But I have some questions. According to this complication, do you consider the use of guided block versus blind block advantageous in cats? It seems contradictory that despite a correct visualization of the structures by ultrasound, brainstem anaesthesia occurred.
It is not entirely clear which is the main hypothesis that the authors indicate to explain the blockade. It would appear to have been an accidental intra-arterial injection. Although intravenous injections may go undetected, it would be unusual for the authors not to detect blood in the aid prior to injection, and it stands to reason that the symptoms would have subsided much faster than in this case. For this reviewer, the duration of the effect suggests that the passage of ropivacaine into the subarachnoid space through contact with the optic nerve sheath could probably be the most probable cause. Or perhaps, a combination of both hypotheses (vascular & subarachnoid) for a high volume for the retrobulbar space of this cat. This possibility due to add to the discussion.
Other specific questions are shown below.
104. Why was the reason to consider that midazolam could be responsible of apnoea? It is surprising that the use of flumazenil allowed the patient to breathe, since midazolam is a drug with little respiratory action. Do you think it was due to the action of the drug, or could the time elapsed since the retrobulbar block influence the response?
107. How was bleeding quantified? 29 ml seems like a very accurate approximation.
Author Response
Thank you very much for the constructive comments.
Please see our response to your comments, written in italics.
This is an interesting case report about the potential brainstem anaesthesia secondary to retrobulbar block in a cat, with a supra-temporal approach. The description of the case is extensive and clear. Although this complication has been described previously, But I have some questions. According to this complication, do you consider the use of guided block versus blind block advantageous in cats? It seems contradictory that despite a correct visualization of the structures by ultrasound, brainstem anaesthesia occurred.
We still believe that ultrasound-guided retrobulbar block is more advantageous compared to the blind technique in cats although we had this complication. In our case, as we mention in the discussion, we didn’t use the Doppler colour flow to visualise better the vessels. Having the option to do that could help to avoid those structures better.
It is not entirely clear which is the main hypothesis that the authors indicate to explain the blockade. It would appear to have been an accidental intra-arterial injection. Although intravenous injections may go undetected, it would be unusual for the authors not to detect blood in the aid prior to injection, and it stands to reason that the symptoms would have subsided much faster than in this case. For this reviewer, the duration of the effect suggests that the passage of ropivacaine into the subarachnoid space through contact with the optic nerve sheath could probably be the most probable cause. Or perhaps, a combination of both hypotheses (vascular & subarachnoid) for a high volume for the retrobulbar space of this cat. This possibility due to add to the discussion.
It is the very quick onset of the symptoms that differentiate this case from the previously reported possible brainstem anaesthesia in a cat. However, we agree that a combination of an intraarterial injection and a sheath penetration cannot be excluded. For this reason, we have altered the discussion and our conclusion.
Other specific questions are shown below.
- Why was the reason to consider that midazolam could be responsible of apnoea? It is surprising that the use of flumazenil allowed the patient to breathe, since midazolam is a drug with little respiratory action. Do you think it was due to the action of the drug, or could the time elapsed since the retrobulbar block influence the response?
We don’t believe that midazolam was responsible for the apnoea as the cat was breathing spontaneously after induction of anaesthesia before the RBB, and the midazolam had been administered IM in the premedication. When the patient was not returning to spontaneous ventilation after the end of anaesthesia, our effort was to reverse any medications that could potentially compromise ventilation. To our surprise the result was immediate. It is interesting that the other reviewer comments that benzodiazepines can cause respiratory depression, especially when used with opioids or when CNS dysfunction is already present. Additionally, she/he provided us with evidence that flumazenil can reduce recovery time from anaesthesia in humans even when benzodiazepines have not been used. We hope that after taking into consideration the suggestions of both of the reviewers we have improved the discussion and have given more food for thought regarding this matter.
- How was bleeding quantified? 29 ml seems like a very accurate approximation.
All gauzes were weighted using a sensitive scale as per our standard protocol to measure blood loss in enucleations.
Round 2
Reviewer 1 Report
Thank you for the changes, great case presentation and discussion